# Benchmarking methods for computing local sensitivities in ordinary differential equation models at dynamic and steady states

**Polina Lakrisenko**[1,3], **Dilan Pathirana**[2], **Daniel Weindl**[1,2], **Jan Hasenauer**[1,2]*

**1** Computational Health Center, Helmholtz Zentrum München Deutsches Forschungszentrum für Gesundheit und Umwelt (GmbH), Neuherberg, Germany, **2** Faculty of Mathematics and Natural Sciences, and the Life and Medical Sciences Institute (LIMES), Rheinische Friedrich-Wilhelms-Universität Bonn, Bonn, Germany, **3** School of Life Sciences, Technische Universität München, Freising, Germany

\* jan.hasenauer@uni-bonn.de

**Data Availability Statement:** All relevant data for this study are publicly available from the Zenodo

## Abstract

Estimating parameters of dynamic models from experimental data is a challenging, and often computationally-demanding task. It requires a large number of model simulations and objective function gradient computations, if gradient-based optimization is used. In many cases, steady-state computation is a part of model simulation, either due to steady-state data or an assumption that the system is at steady state at the initial time point. Various methods are available for steady-state and gradient computation. Yet, the most efficient pair of methods (one for steady states, one for gradients) for a particular model is often not clear. In order to facilitate the selection of methods, we explore six method pairs for computing the steady state and sensitivities at steady state using six real-world problems. The method pairs involve numerical integration or Newton's method to compute the steady-state, and—for both forward and adjoint sensitivity analysis—numerical integration or a tailored method to compute the sensitivities at steady-state. Our evaluation shows that all method pairs provide accurate steady-state and gradient values, and that the two method pairs that combine numerical integration for the steady-state with a tailored method for the sensitivities at steady-state were the most robust, and amongst the most computationally-efficient. We also observed that while Newton's method for steady-state computation yields a substantial speedup compared to numerical integration, it may lead to a large number of simulation failures. Overall, our study provides a concise overview across current methods for computing sensitivities at steady state. While our study shows that there is no universally-best method pair, it also provides guidance to modelers in choosing the right methods for a problem at hand.

## Introduction

At every step of dynamic modeling of biochemical reaction networks, from model formulation and calibration to model quality assessment, one has to choose from a variety of methods and

repository (https://doi.org/10.5281/zenodo.11147151).

**Funding:** This work was supported by the Deutsche Forschungsgemeinschaft (DFG, German Research Foundation, https://www.dfg.de/) under Germany's Excellence Strategy (EXC 2047—390685813, EXC 2151—390873048; J.H.) and under the project IDs 432325352 – SFB 1454 and 443187771 – AMICI (D.P.), by the German Federal Ministry of Education and Research (BMBF, https://www.bmbf.de) within the e:Med funding scheme (junior research alliance PeriNAA, grant no. 01ZX1916A; D.W., P.L.), and by the University of Bonn (https://www.uni-bonn.de, via the Schlegel Professorship of J.H.). The funders had no role in study design, data collection and analysis, decision to publish, or preparation of the manuscript.

**Competing interests:** The authors have declared that no competing interests exist.

approaches [1, 2]. While some decisions can be made relatively quickly based on the scope of the problem and available data, other decisions require an assessment of alternatives on a trial-and-error basis. Therefore, studies that benchmark different algorithms on diverse problems help modelers make informed decisions and recognize potential problems. Many benchmark studies have been published that compare methods available for different modeling steps, such as numerical integration methods for ordinary differential equation (ODE) models [3], computation of derivatives of differential equation solutions [4], solving differential equations and parameter estimation in Julia [5], optimization using different objective functions and approaches to link data to model simulations [6], and for parameter estimation in large kinetic models [7]. Collections of benchmark problems for model simulation or parameter estimation are valuable resources for such method evaluations [8, 9].

Model calibration, i.e. estimation of model parameters from experimental data, is a particularly challenging step that requires a large number of model simulations and, thus, can be computationally demanding [2]. Therefore, for working with larger models in particular, choosing the most efficient model simulation method is important for keeping computations tractable.

In many applications, steady-state computation is required during a model simulation. For example, in a benchmark collection of mathematical models with experimental data curated from the systems biology literature, 22% (7/32) require steady-state computation [9]. One can distinguish two cases: pre-equilibration, where the system is assumed to be in a steady state at the initial time point, then is perturbed and enters a dynamic state; and post-equilibration, which is required if steady-state data is available, where the system is assumed to reach a steady state having started from a dynamic state [10]. These two cases can occur together in the same data set.

Steady-state constraints constitute important prior knowledge, or assumptions, and allow one to reduce the search space during optimization. However, these constraints also pose a challenge, as they often come in a form of nonlinear equalities that can cause efficiency and convergence problems and require special treatment [11–13]. To eliminate steady-state constraints, various approaches have been developed to derive analytical expressions for steady states [13–16]. Furthermore, manifold optimization techniques have been adapted for steady-state constraints [11]. Still, the derivation of analytical expressions for steady states is only applicable to some application problems and manifold optimization techniques can be difficult to tune. Accordingly, most state-of-the-art implementations still rely on the numerical computation of steady states for given parameters. Often, numerical integration is run until a steady state is reached. An alternative is to directly solve for the steady state by using Newton's method (and variations thereof) to find the zeroes of the ODE right-hand-side. However, the choice is non-trivial because Newton's method can be 100 times faster, but numerical simulation is much more robust [12].

As the steady state is usually parameter dependent, multiple methods have also been introduced to compute the sensitivity of the steady state and the objective function gradient. The most widely used approaches for computing local sensitivities include finite differences, automatic differentiation, forward sensitivity analysis (FSA), and adjoint sensitivity analysis (ASA). These methods have been compared for dynamic states [4, 17]. Also, for the computation of steady-state sensitivities, tailored FSA and ASA methods have been implemented that avoid numerical integration and require solving a system of linear equations instead [10, 13]. Yet, a comprehensive comparison of steady-state sensitivity analysis methods and pairs of steady-steady computation and sensitivity analysis schemes has not been performed.

In this study, we consider steady-state and sensitivity-at-steady-state methods that are broadly applicable, i.e. methods that: do not rely on the model structure, are applicable to both pre- and post-equilibration, and work both with steady states alone and in combination with

dynamic states. We study six different pairs of methods and apply them to six real-world problems, and investigate how each method pair affects computation time and failure rate. Overall, we find that Newton's method should be used with caution, as it is less robust than numerical integration. We conclude that the two method pairs that combine numerical integration for the steady state with a tailored method for the sensitivities at steady-state stand out favorably in terms of overall efficiency and robustness.

## Methods

We consider dynamic models of biochemical processes that can be described by systems of ODEs and are defined as initial value problems

$$\dot{\mathbf{x}} = \mathbf{f}(\mathbf{x}(t, \boldsymbol{\theta}, \mathbf{u}), \boldsymbol{\theta}, \mathbf{u}), \ \mathbf{x}(t_0, \boldsymbol{\theta}, \mathbf{u}) = \mathbf{x}_0(\boldsymbol{\theta}, \mathbf{u}), \tag{1}$$

which determine the evolution in time, denoted by $t \in \mathbb{R}$, of a vector of state variables $\mathbf{x}(t, \boldsymbol{\theta}, \mathbf{u}) \in \mathbb{R}^{n_x}$, representing, e.g., abundances of biochemical species. The vector $\boldsymbol{\theta} \in \mathbb{R}^{n_\theta}$ denotes unknown parameters. The vector $\mathbf{u} \in \mathbb{R}^{n_u}$ denotes known inputs that can specify experimental conditions by directly specifying model component values that represent, for example, various external stimuli such as growth factors, receptor ligands, drug administration regimes or changes in cell culture medium composition. The vector field of the ODE model is $\mathbf{f} : \mathbb{R}^{n_x} \times \mathbb{R}^{n_\theta} \times \mathbb{R}^{n_u} \to \mathbb{R}^{n_x}$, which is assumed to be Lipschitz-continuous with respect to $\mathbf{x}$, and the initial condition at $t = t_0$ is $\mathbf{x}_0 : \mathbb{R}^{n_\theta} \times \mathbb{R}^{n_u} \to \mathbb{R}^{n_x}$. Depending on a specific application, domains of $\mathbf{x}$, $\boldsymbol{\theta}$, and $\mathbf{u}$ might be further restricted to ensure plausibility. In systems biology applications, both state variables and unknown parameters are often constrained to be non-negative, as $\mathbf{x}$, for example, might describe species concentrations and $\boldsymbol{\theta}$ might denote reaction rate constants. This restriction, however, is not necessary for the methods described in this study.

The connection between data and the model is provided by the observation map $\mathbf{h} : \mathbb{R}^{n_x} \times \mathbb{R}^{n_\theta} \times \mathbb{R}^{n_u} \to \mathbb{R}^{n_y}$ and the vector of observables,

$$\mathbf{y}(t, \boldsymbol{\theta}, \mathbf{u}) = \mathbf{h}(\mathbf{x}(t, \boldsymbol{\theta}, \mathbf{u}), \boldsymbol{\theta}, \mathbf{u}).$$

The experimental data for these observables are denoted by $\mathcal{D} = \{(t_j, \bar{y}_{ij}) : i = 1, \dots, n_y, j = 1, \dots, n_t\}$, with measurement time points $t_j$ and measurements $\bar{y}_{ij}$.

### Maximum likelihood estimation with forward or adjoint sensitivity analysis

Frequently, the only way to obtain appropriate values for parameters $\boldsymbol{\theta}$ is to estimate them from experimental data $\mathcal{D}$. This can be done, for example, using the maximum likelihood approach, where the likelihood ($\mathcal{L}_{\mathcal{D}}(\boldsymbol{\theta})$) is the probability of the measurements given the model [2]. In the case of independent, normally distributed measurements, the negative log-likelihood is

$$\mathcal{J}(\boldsymbol{\theta}) = -\log \mathcal{L}_{\mathcal{D}}(\boldsymbol{\theta}) = \frac{1}{2} \sum_{i=1}^{n_y} \sum_{j=1}^{n_t} \left( \log \left( 2\pi \sigma_{ij}^2(\boldsymbol{\theta}) \right) + \left( \frac{\bar{y}_{ij} - y_i(t_j, \boldsymbol{\theta}, \mathbf{u})}{\sigma_{ij}(\boldsymbol{\theta})} \right)^2 \right), \tag{2}$$

where $\bar{y}_{ij} = y_i(t_j, \boldsymbol{\theta}, \mathbf{u}) + \varepsilon_{ij}$ are measurements with additive noise $\varepsilon_{ij} \sim \mathcal{N}(0, \sigma_{ij}^2(\boldsymbol{\theta}))$, $\sigma_{ij}(\boldsymbol{\theta})$ are standard deviations, and $y_i(t_j, \boldsymbol{\theta})$ are model simulations.

The maximum likelihood estimate of the parameters is the solution to the optimization problem

$$\boldsymbol{\theta}^* = \arg\min_{\boldsymbol{\theta}} \ \mathcal{J}(\boldsymbol{\theta}).$$

This problem can be efficiently solved using gradient-based global and multi-start optimization methods [1, 7]. These methods require the computation of the objective function gradient, which can be achieved, e.g., by using the finite differences approach, FSA, or ASA. Both FSA and ASA are more reliable and efficient than finite differences, which can be numerically unstable [1, 17]. In the following, we describe both in detail and how they can be modified at steady state.

The gradient of the objective function (2) with respect to the parameters is given by

$$\frac{\partial \mathcal{J}}{\partial \theta_k}\bigg|_{\boldsymbol{\theta}} = \sum_{i=1}^{n_y} \sum_{j=1}^{n_t} \left( \frac{1}{\sigma_{ij}(\boldsymbol{\theta})} \left( 1 - \frac{(\bar{y}_{ij} - y_i(t_j, \boldsymbol{\theta}, \mathbf{u}))^2}{\sigma_{ij}^2(\boldsymbol{\theta})} \right) \frac{\partial \sigma_{ij}}{\partial \theta_k}\bigg|_{\boldsymbol{\theta}} - \frac{(\bar{y}_{ij} - y_i(t_j, \boldsymbol{\theta}, \mathbf{u}))}{\sigma_{ij}^2(\boldsymbol{\theta})} \frac{\partial y_i}{\partial \theta_k}\bigg|_{t_j, \boldsymbol{\theta}, \mathbf{u}} \right),$$

where the sensitivity of the output $y_i$ with respect to parameter $\theta_k$ can be computed by

$$\frac{\partial y_i}{\partial \theta_k}\bigg|_{t, \boldsymbol{\theta}, \mathbf{u}} = \frac{\partial h_i}{\partial \mathbf{x}}\bigg|_{\mathbf{x}(t, \boldsymbol{\theta}, \mathbf{u}), \boldsymbol{\theta}, \mathbf{u}} \frac{\partial \mathbf{x}}{\partial \theta_k}\bigg|_{t, \boldsymbol{\theta}, \mathbf{u}} + \frac{\partial h_i}{\partial \theta_k}\bigg|_{\mathbf{x}(t, \boldsymbol{\theta}, \mathbf{u}), \boldsymbol{\theta}, \mathbf{u}}. \tag{3}$$

The term $\frac{\partial \mathbf{x}}{\partial \theta_k}$ is the vector of state sensitivities with respect to parameter $\theta_k$, also denoted by $\mathbf{s}_k^{\mathbf{x}}$. The state sensitivities can be tricky to compute, as $\mathbf{x}$ is only given as the solution of the ODE (1).

One can calculate state sensitivities by using FSA. By differentiating the equations (1) with respect to $\boldsymbol{\theta}$, one can get an ODE system governing state sensitivities $\mathbf{s}_k^{\mathbf{x}}$:

$$\dot{\mathbf{s}}_k^{\mathbf{x}} = \frac{\partial \mathbf{f}}{\partial \mathbf{x}}\bigg|_{\mathbf{x}(t, \boldsymbol{\theta}, \mathbf{u}), \boldsymbol{\theta}, \mathbf{u}} \mathbf{s}_k^{\mathbf{x}}(t, \boldsymbol{\theta}, \mathbf{u}) + \frac{\partial \mathbf{f}}{\partial \theta_k}\bigg|_{\mathbf{x}(t, \boldsymbol{\theta}, \mathbf{u}), \boldsymbol{\theta}, \mathbf{u}}, \ \text{with } \mathbf{s}_k^{\mathbf{x}}(t_0, \boldsymbol{\theta}, \mathbf{u}) = \frac{\partial \mathbf{x}_0}{\partial \theta_k}\bigg|_{\boldsymbol{\theta}, \mathbf{u}}, \tag{4}$$

where $k = 1, \ldots, n_\theta$ and $\frac{\partial \mathbf{f}}{\partial \mathbf{x}}$ represents the Jacobian of the system (1)

$$\frac{\partial \mathbf{f}}{\partial \mathbf{x}}\bigg|_{\mathbf{x}(t, \boldsymbol{\theta}, \mathbf{u}), \boldsymbol{\theta}, \mathbf{u}} = \begin{bmatrix} \frac{\partial f_1}{\partial x_1} & \cdots & \frac{\partial f_1}{\partial x_{n_x}} \\ \vdots & \ddots & \vdots \\ \frac{\partial f_{n_x}}{\partial x_1} & \cdots & \frac{\partial f_{n_x}}{\partial x_{n_x}} \end{bmatrix}_{\mathbf{x}(t, \boldsymbol{\theta}, \mathbf{u}), \boldsymbol{\theta}, \mathbf{u}}.$$

Solving this ODE system yields state sensitivities, from which output sensitivities $\frac{\partial y_i}{\partial \theta_k}$ can be easily calculated by (3). Numerical integration of the state sensitivities ODEs (4) is often coupled with integration of the system (1), which improves computational efficiency [18, 19].

An alternative to FSA is ASA, which allows for the calculation of the objective function gradient while avoiding calculation of output sensitivities [17]. It is achieved by introducing the adjoint state $\mathbf{p}(t, \boldsymbol{\theta}, \mathbf{u}) : [t_0, t_{n_t}] \times \mathbb{R}^{n_\theta} \times \mathbb{R}^{n_u} \to \mathbb{R}^{n_x}$, such that $\forall j = n_t, \ldots, 1$, $\mathbf{p}$ on interval $(t_{j-1}, t_j]$ satisfies the backward differential equation

$$\dot{\mathbf{p}}(t, \boldsymbol{\theta}, \mathbf{u}) = -\frac{\partial \mathbf{f}}{\partial \mathbf{x}}\bigg|_{\mathbf{x}(t, \boldsymbol{\theta}, \mathbf{u}), \boldsymbol{\theta}, \mathbf{u}}^T \mathbf{p}(t, \boldsymbol{\theta}, \mathbf{u}), \tag{5}$$

with boundary values

$$\mathbf{p}(t_j, \boldsymbol{\theta}, \mathbf{u}) = \lim_{t \to t_j^+} \mathbf{p}(t, \boldsymbol{\theta}, \mathbf{u}) + \sum_{i=1}^{n_y} \frac{\partial h_i}{\partial \mathbf{x}}\bigg|^T_{\mathbf{x}(t_j, \boldsymbol{\theta}, \mathbf{u}), \boldsymbol{\theta}, \mathbf{u}} \frac{(\bar{y}_{ij} - y_i(t_j, \boldsymbol{\theta}, \mathbf{u}))}{\sigma_{ij}^2(\boldsymbol{\theta})}, \quad \text{and}$$

$$\lim_{t \to t_{n_t}^+} \mathbf{p}(t, \boldsymbol{\theta}, \mathbf{u}) = 0.$$

The adjoint state is used to reformulate the gradient

$$\begin{aligned}
\frac{\partial \mathcal{J}}{\partial \theta_k}\bigg|_{\boldsymbol{\theta}} &= \sum_{i=1}^{n_y}\sum_{j=1}^{n_t}\left(\frac{1}{\sigma_{ij}(\boldsymbol{\theta})}\left(1 - \frac{(\bar{y}_{ij} - y_i(t_j, \boldsymbol{\theta}, \mathbf{u}))^2}{\sigma_{ij}^2(\boldsymbol{\theta})}\right)\frac{\partial \sigma_{ij}}{\partial \theta_k}\bigg|_{\boldsymbol{\theta}}\right) \\
&\quad - \sum_{i=1}^{n_y}\sum_{j=1}^{n_t}\frac{(\bar{y}_{ij} - y_i(t_j, \boldsymbol{\theta}, \mathbf{u}))}{\sigma_{ij}^2(\boldsymbol{\theta})}\frac{\partial h_i}{\partial \theta_k}\bigg|_{\mathbf{x}(t, \boldsymbol{\theta}, \mathbf{u}), \boldsymbol{\theta}, \mathbf{u}} \\
&\quad - \int_{t_0}^{t_{n_t}} \mathbf{p}(t, \boldsymbol{\theta}, \mathbf{u})^T \frac{\partial \mathbf{f}}{\partial \theta_k}\bigg|_{\mathbf{x}(t, \boldsymbol{\theta}, \mathbf{u}), \boldsymbol{\theta}, \mathbf{u}} dt - \mathbf{p}(t_0, \boldsymbol{\theta}, \mathbf{u})^T \frac{\partial \mathbf{x}_0}{\partial \theta_k}\bigg|_{\boldsymbol{\theta}, \mathbf{u}},
\end{aligned} \tag{6}$$

in which $\partial \mathbf{x}_0 / \partial \theta_k$ denotes the sensitivity of the initial state with respect to parameter $\theta_k$. As was shown in [17], this approach is computationally more efficient than finite differences or FSA for models with high numbers of state variables or parameters.

## Steady-state computation

In this study, we focus on cases in which evaluating the objective function involves the computation of a parameter-dependent steady state $\mathbf{x}^*(\boldsymbol{\theta}, \mathbf{u})$:

$$\mathbf{x}^*(\boldsymbol{\theta}, \mathbf{u}) = \lim_{t \to \infty} \mathbf{x}(t, \mathbf{x}_0(\boldsymbol{\theta}, \mathbf{u}), \boldsymbol{\theta}, \mathbf{u}). \tag{7}$$

This is rather common, as before and after perturbations the biological systems are often in an asymptotically-stable steady state or approach one. Exceptions are oscillating and chaotic systems.

There are two typical cases [10]:

1. **pre-equilibration**, where at $t = t_0$, $\mathbf{x}$ is the steady state associated with input $\mathbf{u}^e$; and

2. **post-equilibration**, where as $t \to \infty$, $\mathbf{x}$ approaches the steady state $\mathbf{x}^*(\boldsymbol{\theta}, \mathbf{u})$ associated with input $\mathbf{u}$.

In both cases, according to Picard-Lindelöf theorem, the system (1) has a unique steady state for each initial condition $(\mathbf{x}_0(\boldsymbol{\theta}, \mathbf{u}))$ [20].

To compute the steady state of the ODE system (1), one must solve for $\mathbf{x}$ the equation

$$\mathbf{f}(\mathbf{x}(t, \boldsymbol{\theta}, \mathbf{u}), \boldsymbol{\theta}, \mathbf{u}) = 0,$$

which is generally nonlinear and a closed-form solution is not available. The most straightforward numerical approach is to integrate the system of ODEs until time derivatives $\dot{\mathbf{x}}$ become sufficiently small. For example, the integration can be performed until condition

$$\sqrt{\frac{1}{n_x}\sum_{i=1}^{n_x}(\dot{x}_i w_i)^2} < 1, \quad \text{where } w_i = \frac{1}{\text{rtol} \cdot x_i + \text{atol}}, \tag{8}$$

is fulfilled, where "rtol" and "atol" denote relative and absolute tolerances, respectively.

Another well-known approach is Newton's method, where the approximation $\mathbf{x}^l$ of the solution is computed iteratively by formula

$$\mathbf{x}^{l+1} = \mathbf{x}^l - \left( \frac{\partial \mathbf{f}}{\partial \mathbf{x}} \Big|_{(\mathbf{x}^l, \boldsymbol{\theta}, \mathbf{u})} \right)^{-1} \mathbf{f}(\mathbf{x}^l, \boldsymbol{\theta}, \mathbf{u}), \; l = 0, 1, \ldots,$$

until convergence criterion, e.g. (8), is fulfilled. The method is very sensitive to the initial guess $\mathbf{x}^0$ and may not converge if started too far away from the solution. One can extend the radius of convergence by modifying the step length:

$$\mathbf{x}^{l+1} = \mathbf{x}^l - \gamma \left( \frac{\partial \mathbf{f}}{\partial \mathbf{x}} \Big|_{(\mathbf{x}^l, \boldsymbol{\theta}, \mathbf{u})} \right)^{-1} \mathbf{f}(\mathbf{x}^l, \boldsymbol{\theta}, \mathbf{u}), \tag{9}$$

where $\gamma \leq 1$ [12]. The factor $\gamma$ is increased if the error ($\sqrt{\frac{1}{n_x} \sum_{i=1}^{n_x} (\dot{x}_i w_i)^2}$ for condition (8)) is reduced and decreased otherwise. When successful, Newton's method can be orders of magnitude faster than numerical integration [12].

## Objective function gradient computation at steady state

If either pre- or post-equilibration is required, it affects objective function gradient computation:

1. **Pre-equilibration.** In this case, the initial condition is defined as the steady state corresponding to a pre-equilibration input $\mathbf{u}^e$: $\mathbf{x}_0(\boldsymbol{\theta}, \mathbf{u}) = \mathbf{x}^*(\boldsymbol{\theta}, \mathbf{u}^e)$, where $\mathbf{x}^*$ is the steady state of

$$\dot{\mathbf{x}} = \mathbf{f}(\mathbf{x}(t, \boldsymbol{\theta}, \mathbf{u}^e), \boldsymbol{\theta}, \mathbf{u}^e), \; \mathbf{x}(t_0, \boldsymbol{\theta}) = \mathbf{x}_0(\boldsymbol{\theta}, \mathbf{u}^e).$$

Here, $\mathbf{x}^*$ affects $\mathbf{x}(t_j, \boldsymbol{\theta}, \mathbf{u})$ and, consequently, the simulated observable values $\mathbf{y}(t_j, \boldsymbol{\theta}, \mathbf{u})$. Moreover, it influences sensitivity computation both for FSA and ASA, namely, the initial condition in (4) and the last term in (6).

2. **Post-equilibration.** In this case one needs to account for steady-state measurements

$$\bar{y}_{i*} = y_{i*}(\boldsymbol{\theta}, \mathbf{u}) + \varepsilon_{i*} \quad \text{with } \varepsilon_{i*} \sim \mathcal{N}(0, \sigma_{i*}^2(\boldsymbol{\theta})),$$

in which $y_{i*}(\boldsymbol{\theta}, \mathbf{u}) = h_i(\mathbf{x}^*(\boldsymbol{\theta}, \mathbf{u}), \boldsymbol{\theta}, \mathbf{u})$ denotes the values of the observable $y_i$ at the steady state (7), and $\sigma_{i*}^2(\boldsymbol{\theta}) \in \mathbb{R}_+$ denotes the noise variance. In practice, post-equilibration measurements arise when measuring a real system that appears to have reached a steady state, e.g. long after the last input change occurred. It is possible that both steady-state and time-course measurements are available. Therefore, in this study we consider the most general case.

The steady-state measurements need to be included in the objective function,

$$\mathcal{J}(\boldsymbol{\theta}) = -\log \mathcal{L}_{\mathcal{D}}(\boldsymbol{\theta}) = \frac{1}{2} \sum_{i=1}^{n_y} \sum_{j=1}^{n_t} \left( \log \left( 2\pi \sigma_{ij}^2(\boldsymbol{\theta}) \right) + \left( \frac{\bar{y}_{ij} - y_i(t_j, \boldsymbol{\theta}, \mathbf{u})}{\sigma_{ij}(\boldsymbol{\theta})} \right)^2 \right)$$

$$+ \frac{1}{2} \sum_{i=1}^{n_y} \left( \log \left( 2\pi \sigma_{i*}^2(\boldsymbol{\theta}) \right) + \left( \frac{\bar{y}_{i*} - y_{i*}(\boldsymbol{\theta}, \mathbf{u})}{\sigma_{i*}(\boldsymbol{\theta})} \right)^2 \right),$$

and its gradient,

$$
\begin{aligned}
\left.\frac{\partial \mathcal{J}}{\partial \theta_k}\right|_{\boldsymbol{\theta}} &= \sum_{i=1}^{n_y}\sum_{j=1}^{n_t}\left(\frac{1}{\sigma_{ij}(\boldsymbol{\theta})}\left(1 - \frac{(\bar{y}_{ij} - y_i(t_j,\boldsymbol{\theta},\mathbf{u}))^2}{\sigma_{ij}^2(\boldsymbol{\theta})}\right)\left.\frac{\partial \sigma_{ij}}{\partial \theta_k}\right|_{\boldsymbol{\theta}} - \frac{(\bar{y}_{ij} - y_i(t_j,\boldsymbol{\theta},\mathbf{u}))}{\sigma_{ij}^2(\boldsymbol{\theta})}\left.\frac{\partial y_i}{\partial \theta_k}\right|_{t_j,\boldsymbol{\theta},\mathbf{u}}\right) \\
&\quad - \sum_{i=1}^{n_y}\left(\frac{(\bar{y}_{i*} - y_{i*}(\boldsymbol{\theta},\mathbf{u}))^2}{\sigma_{i*}^3(\boldsymbol{\theta})}\left.\frac{\partial \sigma_{i*}}{\partial \theta_k}\right|_{\boldsymbol{\theta}} + \frac{(\bar{y}_{i*} - y_{i*}(\boldsymbol{\theta},\mathbf{u}))}{\sigma_{i*}^2(\boldsymbol{\theta})}\left.\frac{\partial y_{i*}}{\partial \theta_k}\right|_{\boldsymbol{\theta},\mathbf{u}}\right).
\end{aligned}
$$

## Tailored methods for handling steady states

The steady-state condition facilitates alternative approaches for sensitivity-at-steady-state computation. In this section we describe how it can be used to simplify gradient computation both in FSA and ASA.

As $\dot{\mathbf{s}}_k^{\mathbf{x}}|_{\mathbf{x}=\mathbf{x}^*} = 0$ for all $k = 1, \ldots, n_\theta$, the forward sensitivities ODE system (4) simplifies to

$$
\left.\frac{\partial \mathbf{f}}{\partial \mathbf{x}}\right|_{\mathbf{x}^*(\boldsymbol{\theta},\mathbf{u}),\boldsymbol{\theta},\mathbf{u}}\mathbf{s}_k^{\mathbf{x}}|_{\mathbf{x}^*(\boldsymbol{\theta},\mathbf{u}),\boldsymbol{\theta},\mathbf{u}} = -\left.\frac{\partial \mathbf{f}}{\partial \theta_k}\right|_{\mathbf{x}^*(\boldsymbol{\theta},\mathbf{u}),\boldsymbol{\theta},\mathbf{u}}, \quad k = 1, \ldots, n_\theta. \tag{10}
$$

Therefore, if the Jacobian of the system (1) has full rank, the system of $n_x n_\theta$ FSA ODEs (4) simplifies to a system of the same number of linear algebraic equations [11, 13]. This approach is applicable to both the pre- and post-equilibration cases.

If the adjoint sensitivities approach is used, computations can be simplified as well. In the pre-equilibration case, one can extend ASA to the pre-equilibration time interval $[-t', t_0]$, where $-t'$ is such that the steady state has already been achieved. Then objective function gradient is given by

$$
\begin{aligned}
\left.\frac{\partial \mathcal{J}}{\partial \theta_k}\right|_{\boldsymbol{\theta}} &= \sum_{i=1}^{n_y}\sum_{j=1}^{n_t}\left(\frac{1}{\sigma_{ij}(\boldsymbol{\theta})}\left(1 - \frac{(\bar{y}_{ij} - y_i(t_j,\boldsymbol{\theta},\mathbf{u}))^2}{\sigma_{ij}^2(\boldsymbol{\theta})}\right)\left.\frac{\partial \sigma_{ij}}{\partial \theta_k}\right|_{\boldsymbol{\theta}}\right) \\
&\quad - \sum_{i=1}^{n_y}\sum_{j=1}^{n_t}\frac{(\bar{y}_{ij} - y_i(t_j,\boldsymbol{\theta},\mathbf{u}))}{\sigma_{ij}^2(\boldsymbol{\theta})}\left.\frac{\partial h_i}{\partial \theta_k}\right|_{\mathbf{x}(t_j,\boldsymbol{\theta},\mathbf{u}),\boldsymbol{\theta},\mathbf{u}} \\
&\quad - \int_{-t'}^{t_0}\mathbf{p}(t,\boldsymbol{\theta},\mathbf{u}^e)^T\left.\frac{\partial \mathbf{f}}{\partial \theta_k}\right|_{\mathbf{x}_0,\boldsymbol{\theta},\mathbf{u}^e}dt \\
&\quad - \int_{t_0}^{t_{n_t}}\mathbf{p}(t,\boldsymbol{\theta},\mathbf{u})^T\left.\frac{\partial \mathbf{f}}{\partial \theta_k}\right|_{\mathbf{x}(t,\boldsymbol{\theta},\mathbf{u}),\boldsymbol{\theta},\mathbf{u}}dt - \mathbf{p}(-t',\boldsymbol{\theta},\mathbf{u}^e)^T\left.\frac{\partial \mathbf{x}_0}{\partial \theta_k}\right|_{\boldsymbol{\theta},\mathbf{u}}.
\end{aligned} \tag{11}
$$

On this time interval the system (5) is a linear ODE system with constant matrix. At $t = -t'$ this system is at steady state $\mathbf{p} = \mathbf{0}$, therefore, the last term in (11) vanishes. The third term in (11) reduces to a matrix-vector product

$$
\mathbf{P}_{\text{integral}} \cdot \left.\frac{\partial \mathbf{f}}{\partial \theta_k}\right|_{\mathbf{x}^*(\boldsymbol{\theta},\mathbf{u}^e),\boldsymbol{\theta},\mathbf{u}^e},
$$

where $\mathbf{p}_{\text{integral}}$ can be computed as the solution of the system of linear algebraic equations

$$\left.\frac{\partial \mathbf{f}}{\partial \mathbf{x}}\right|_{\mathbf{x}^*(\theta,\mathbf{u}^e),\theta,\mathbf{u}^e}^{T} \mathbf{p}_{\text{integral}} = -\mathbf{p}(t_0,\theta,\mathbf{u}). \tag{12}$$

In the post-equilibration case the gradient is given by

$$
\begin{aligned}
\left.\frac{\partial \mathcal{J}}{\partial \theta_k}\right|_{\theta} &= \sum_{i=1}^{n_y}\sum_{j=1}^{n_t}\left(\frac{1}{\sigma_{ij}(\theta)}\left(1 - \frac{(\bar{y}_{ij} - y_i(t_j,\theta,\mathbf{u}))^2}{\sigma_{ij}^2(\theta)}\right)\left.\frac{\partial \sigma_{ij}}{\partial \theta_k}\right|_{\theta}\right) \\
&\quad - \sum_{i=1}^{n_y}\left(\frac{(\bar{y}_{i*} - y_{i*}(\theta,\mathbf{u}))^2}{\sigma_{i*}^3(\theta)}\left.\frac{\partial \sigma_{i*}}{\partial \theta_k}\right|_{\theta}\right) \\
&\quad - \sum_{i=1}^{n_y}\sum_{j=1}^{n_t}\frac{(\bar{y}_{ij} - y_i(t_j,\theta,\mathbf{u}))}{\sigma_{ij}^2(\theta)}\left.\frac{\partial h_i}{\partial \theta_k}\right|_{\mathbf{x}(t,\theta,\mathbf{u}),\theta,\mathbf{u}} - \sum_{i=1}^{n_y}\frac{(\bar{y}_{i*} - y_{i*}(\theta,\mathbf{u}))}{\sigma_{i*}^2(\theta)}\left.\frac{\partial h_i}{\partial \theta_k}\right|_{\mathbf{x}^*(\theta,\mathbf{u}),\theta,\mathbf{u}} \\
&\quad - \int_{t_0}^{t_{n_t}}\mathbf{p}(t,\theta,\mathbf{u})^T\left.\frac{\partial \mathbf{f}}{\partial \theta_k}\right|_{\mathbf{x}(t,\theta,\mathbf{u}),\theta,\mathbf{u}}dt \\
&\quad - \int_{t_{n_t}}^{t''}\mathbf{p}(t,\theta,\mathbf{u})^T\left.\frac{\partial \mathbf{f}}{\partial \theta_k}\right|_{\mathbf{x}(t,\theta,\mathbf{u}),\theta,\mathbf{u}}dt - \mathbf{p}(t_0,\theta,\mathbf{u})^T\left.\frac{\partial \mathbf{x}_0}{\partial \theta_k}\right|_{\theta,\mathbf{u}},
\end{aligned}
\tag{13}
$$

where $t''$ is a time point at which time derivatives $\dot{\mathbf{x}}$ are negligible and $\mathbf{x}(t)$ is a good approximation of the steady-state $\mathbf{x}^*$.

The sixth term in (13) reduces to a matrix-vector product

$$\mathbf{p}_{\text{integral}} \cdot \left.\frac{\partial \mathbf{f}}{\partial \theta_k}\right|_{\mathbf{x}^*(\theta,\mathbf{u}),\theta,\mathbf{u}},$$

where $\mathbf{p}_{\text{integral}}$ can be computed as the solution to

$$\left.\frac{\partial \mathbf{f}}{\partial \mathbf{x}}\right|_{\mathbf{x}^*(\theta,\mathbf{u}),\theta,\mathbf{u}}^{T} \mathbf{p}_{\text{integral}} = -\mathbf{p}(t'',\theta,\mathbf{u}). \tag{14}$$

The tailored ASA approach for both pre-equilibration and post-equilibration is described in more detail in [10].

One should note that, if the (transposed) Jacobian is not full rank, the tailored approaches are not applicable. In this case, none of the systems (10), (12) and (14) has a unique solution and standard integration must be carried out. A possible reason for a singular Jacobian is the presence of conserved quantities in the model [21]. Various algorithms are available for identification of conserved quantities that facilitate applicability of the tailored methods [22–24].

## Implementation

The six test problems (Table 1) were downloaded from the PEtab benchmark collection [25]. Model simulations and sensitivity computations were performed using AMICI [26]. For numerical integration, an algorithm based on the backward-differentiation formula (BDF) was used. Optimization was performed using the Fides trust-region optimizer [27] via pyPESTO [28]. Simulator options and further details are provided in Section "Implementation" of the S1 Text.

**Table 1. Overview of the models and optimization problems considered in this study.** Here, $n_x$ is the number of state variables, $n_\theta$ is the number of unknown parameters, $n_{\bar{y}}$ is the number of data points, $n_u$ is the number of inputs **u** and "**x\*type**" is the equilibration type.

| Problem | $n_x$ | $n_\theta$ | $n_{\bar{y}}$ | $n_u$ | x*type | Description | Ref. |
|---|---|---|---|---|---|---|---|
| Blasi | 15 | 9 | 252 | 1 | post | The model describes acetylation of the histone H4 *N*-terminal tail domain. Contains 1 conserved quantity that is automatically removed by AMICI, which means that $n_x$ was reduced from 16 to 15. | [29] |
| Brännmark | 6 | 22 | 43 | 8 | pre | The model describes early insulin signalling. Contains 3 conserved quantities that were removed prior to the analysis to ensure a full-rank Jacobian. Consequently, the $n_x$ was reduced from 9 to 6. | [31] |
| Fröhlich | 1396 | 4088 | 143 | 143 | post | The model describes major cancer-associated signaling pathways and facilitates analysis of various cancer types and drug treatments. In this study we used only a subset of data used in the original publication to reduce computation time required for model simulation and thereby optimization too. Specifically, we only used the control conditions, which are 143 out of the total 5281 inputs **u**. Therefore, the number of inputs $n_u$ as well as the number of data points used in this study is equal to 143. | [32] |
| Isensee | 18 | 44 | 216 | 58 | pre | The model describes the activity, in response to various treatments, of the cAMP-dependent protein kinase A, which is crucial for pain sensitization and other biological functions. We used only a subset of data, specifically, 58 out of the total 123 inputs **u** in the original publication, which corresponds to 216 data points out of total 687. The model contains 7 conserved quantities that were removed prior to the analysis to ensure a full-rank Jacobian, which means that $n_x$ was reduced from 25 to 18. | [33] |
| Weber | 7 | 36 | 135 | 2 | pre | The model describes phosphorylation-dependent CERT protein-mediated regulation of ceramide transfer between endoplasmic reticulum and Golgi membranes. | [34] |
| Zheng | 15 | 46 | 60 | 1 | pre | The model describes histone methylation dynamics. | [30] |

## Results

The combination of computing steady states by numerical integration or by Newton's method with FSA, ASA, or their tailored-to-steady-state versions gives rise to eight different method pairs (Fig 1). In this study, we consider the six pairs that are feasible and meaningful (in the following, we use a shorthand notation to refer to the method pairs, where each method pair is represented by a tuple of steady-state and sensitivities-at-steady-state method, $\int$ signifies numerical integration, whereas $\phi$ signifies solving an algebraic equation):

- numerical integration for both steady-state and sensitivities computation ($\langle \int x, \int \mathcal{J}_{\theta,\mathrm{FSA}} \rangle$ and $\langle \int x, \int \mathcal{J}_{\theta,\mathrm{ASA}} \rangle$);

- numerical integration for steady-state computation combined with the sensitivities computation approach tailored to steady-state case ($\langle \int x, \phi \mathcal{J}_{\theta,\mathrm{FSA}} \rangle$ and $\langle \int x, \phi \mathcal{J}_{\theta,\mathrm{ASA}} \rangle$); and

- Newton's method for steady-state computation combined with the sensitivities computation approach tailored to steady-state case ($\langle \phi x, \phi \mathcal{J}_{\theta,\mathrm{FSA}} \rangle$ and $\langle \phi x, \phi \mathcal{J}_{\theta,\mathrm{ASA}} \rangle$).

Note that with FSA, it is not possible to use Newton's method for steady-state computation together with numerical integration approach for sensitivities, as numerical integration of the state sensitivities ODEs (4) [18] is coupled with integration of the system (1) [18]. While the ASA variant of this method pair is feasible, it is not practical, as the tailored method for sensitivities is always applicable when Newton's method can be used for steady-state computation. Therefore, we do not consider these two method pairs in this study.

As it is not immediately clear which of the six method pairs are best in practice, we assess their performance on a selection of real-world problems of varying complexity that required steady-state computation. These models describe epigenetic [29, 30] and signal transduction processes [31–34]. Four of the problems possess pre-equilibration constraints and two possess post-equilibration constraints. The number of states ranges from 6 to 1396 and the number of parameters from 9 to 4088. For details we refer to the information on Table 1. The original

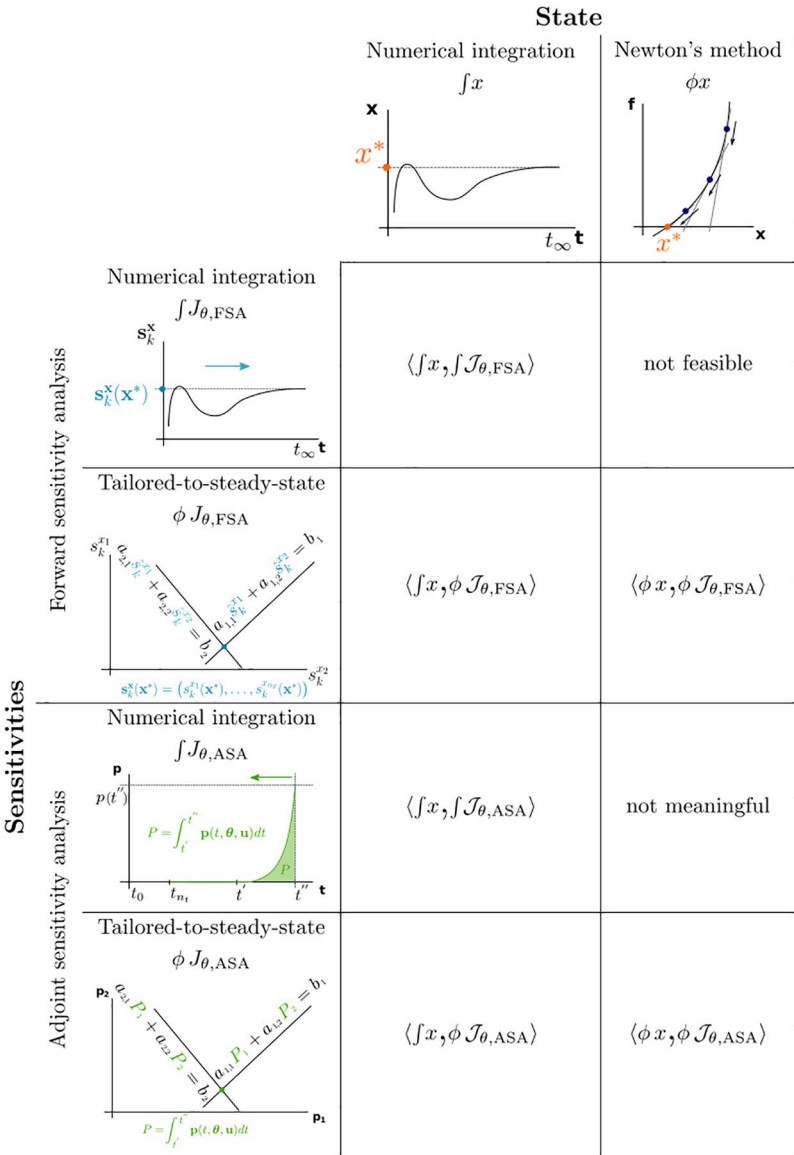

**Fig 1. Method pairs considered in this study.** Columns correspond to two methods for steady-state computation: numerical integration and Newton's method. Rows correspond to different approaches for sensitivities computation at steady state: both for forward sensitivity analysis and adjoint sensitivity analysis one can either use numerical integration or an approach tailored to the steady-state case that only requires solving a linear system of equations. Two method pairs are not feasible or not meaningful (see main text). Each method pair is represented by a tuple of steady-state and sensitivities-at-steady-state method, $\int$ signifies numerical integration, whereas $\phi$ signifies solving an algebraic equation.

`Blasi`, `Brännmark` and `Isensee` models contain conserved quantities that lead to a rank-deficient Jacobian, which is not compatible with the Newton's method and the tailored sensitivities-at-steady-state methods. The conserved quantities were removed prior to the analysis details are given in Section "Conserved quantities" of the S1 Text.

We apply the six method pairs on these six real-world problems to assess robustness, accuracy, and speed, which are crucial for gradient-based parameter estimation.

## Numerical integration is more robust than Newton's method

Model analysis, parameter estimation and related tasks require robust model evaluations. To assess whether this is achieved by the different pairs of steady-state and sensitivity-at-steady-state computation methods, we evaluated their failure rates. Therefore, for each model, we sampled 1000 parameter vectors log-uniformly within the parameter bounds specified in the PEtab files, then performed simulation and gradient computation with each vector. Only for the Fröhlich model the $\langle \int x, \int \mathcal{J}_{\theta,\text{FSA}} \rangle$ case was not considered since it has already been shown to be computationally prohibitive [32]. We consider a model simulation for a given parameter vector as failed if for any input **u** the simulation (a) fails due to any numerical errors, or (b) returns negative state variables in the solution. The latter accounts for the fact that the state variables of all six real-world problems represent physically non-negative quantities.

We observed a broad range of failure rates. For the Blasi and Zheng models, we did not encounter any failures for the 1000 parameter vectors. For the Brännmark, Isensee and Weber models, the failure rate for the six method pairs ranged from 31.5% with $\langle \int x, \int \mathcal{J}_{\theta,\text{ASA}} \rangle$ and $\langle \int x, \phi \mathcal{J}_{\theta,\text{ASA}} \rangle$ to 48.8% with $\langle \phi x, \phi \mathcal{J}_{\theta,\text{FSA}} \rangle$ (Fig 2a), from 7.7% with $\langle \int x, \phi \mathcal{J}_{\theta,\text{FSA}} \rangle$ to 54.3% with $\langle \phi x, \phi \mathcal{J}_{\theta,\text{ASA}} \rangle$ (Fig 2c), and from 22.4% with $\langle \int x, \phi \mathcal{J}_{\theta,\text{ASA}} \rangle$ to 48.3% with $\langle \phi x, \phi \mathcal{J}_{\theta,\text{FSA}} \rangle$ (Fig 2d), respectively. The experimental conditions for the Weber model are described using a discontinuous function, which caused numerical problems in about 17% of the simulations for each of the six method pairs (S1 Fig in S1 Text). With the Brännmark model, we observed that for about 30% of sampled parameters simulations failed independently of the applied method (S1 Fig in S1 Text). This suggests that the model does not exhibit a steady state for those parameter sets. For the Fröhlich model, $\langle \phi x, \phi \mathcal{J}_{\theta,\text{FSA}} \rangle$ and $\langle \phi x, \phi \mathcal{J}_{\theta,\text{ASA}} \rangle$ failed for 100% of the parameter vectors, while the three remaining pairs had a failure rate less than 2.5% percent (Fig 2b).

The reason for the high failure rate with Newton's method varied between models. The most common reason for all models is that the factor $\gamma$ in (9) reached a lower bound while the

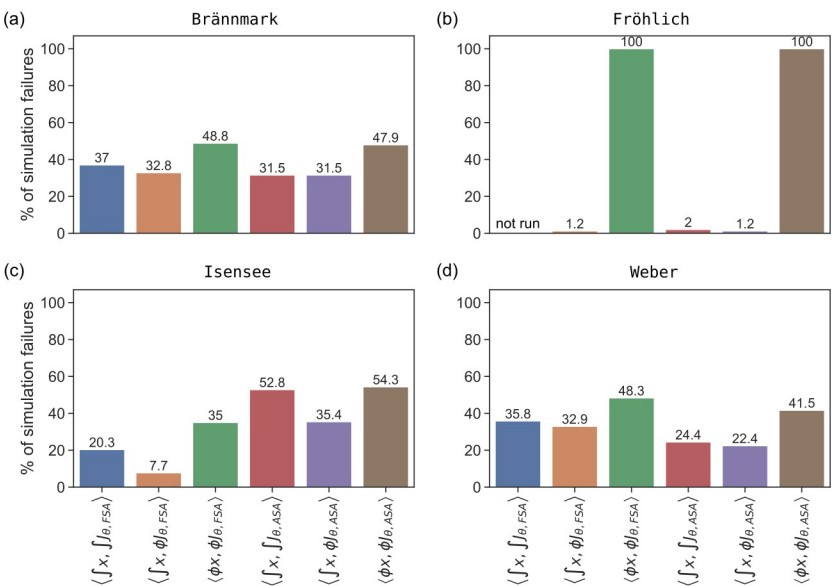

**Fig 2. Failure rates for different method pairs based on model simulations with 1000 randomly sampled parameter vectors.** We did not encounter any failures for the Blasi and Zheng models.

condition (8) was not fulfilled, i.e. the steady state could not be computed because the initial guess was too far away from it, and even the modification described in (9) could not resolve the issue. For the `Fröhlich` model, Newton's method failed with every parameter vector due to numerical errors in at least one of the $n_u$ steady-state computations. Indeed, only 6.6% (9,454/143,000) of the attempts to compute a steady state using $\langle \phi x, \phi \mathcal{J}_{\theta,\text{FSA}} \rangle$ and $\langle \phi x, \phi \mathcal{J}_{\theta,\text{ASA}} \rangle$ did not have numerical errors. The hypothesis that the failure rate for Newton's method depends on the number of state variables could not be confirmed based on the given set of problems—both the `Fröhlich` model with a large number of state variables and the `Brännmark` model with very few state variables exhibited high failure rates. For the `Brännmark`, `Isensee` and `Weber` models, some steady-state concentrations computed with Newton's method had negative values, some of which were several orders of magnitude away from zero, which is not biologically feasible. This issue did not occur when numerical integration was used. For example, for the `Fröhlich` model, concentrations computed with Newton's method had negative values for most inputs **u** simulations that did not have any numerical issues (9,437/9,454). In other words, only 0.01% of inputs **u** (17/143,000) could be simulated successfully with method pairs $\langle \phi x, \phi \mathcal{J}_{\theta,\text{FSA}} \rangle$ and $\langle \phi x, \phi \mathcal{J}_{\theta,\text{ASA}} \rangle$.

In summary, this assessment revealed that numerical simulation allows for a higher fraction of successful simulations than Newton's method.

## All method pairs provide accurate steady-state and gradient computation

Given that a computation is successful, the most important factor is accuracy. To assess the accuracy, we compared the steady-state values obtained by applying the method pairs with the sampled parameter vectors.

For all successful simulations, the steady-state values computed using either numerical integration or Newton's method were similar (S2, S3 Figs in S1 Text). Having confirmed that the different methods yielded approximately equal steady states, we assessed the agreement of computed objective function gradient values. For all models, the objective function gradient values computed using different approaches were similar (Fig 3 and S4 Fig in S1 Text).

In summary, this assessment revealed that for successful simulations the results for different method pairs are highly comparable. We generally saw improved accuracy (increased similarity between method pairs) with stricter simulation tolerances. The minor differences between method pairs seen in Fig 3 may be resolved by tolerance tuning.

## Using tailored sensitivity-at-steady-state methods significantly reduces simulation time

After ensuring the accuracy of the steady-state and gradient values, we assessed the computation time for different method pairs. For all models, the total simulation time is comprised of equilibration time (pre- or post-), including sensitivities computation, and some overhead (Fig 4a). Additionally, the `Brännmark`, `Isensee`, `Weber` and `Zheng` models had dynamic simulation time, because these problems also include dynamic-state measurements. Therefore, we considered again the successful simulations for the previously sampled 1000 parameter vectors, for which we recorded the total equilibration time, and the overall total simulation time (Fig 4b).

The method pairs considered in this work address equilibration. Therefore, the effect of the method pairs on total equilibration time is substantial (Fig 4b, right half of the violins), but it doesn't always translate to the total simulation time (Fig 4b, left half of the violins). The reason is that the difference in the considered method pairs relates only to equilibration time, but not the dynamic simulation, which is a part of model simulations for the `Brännmark`,

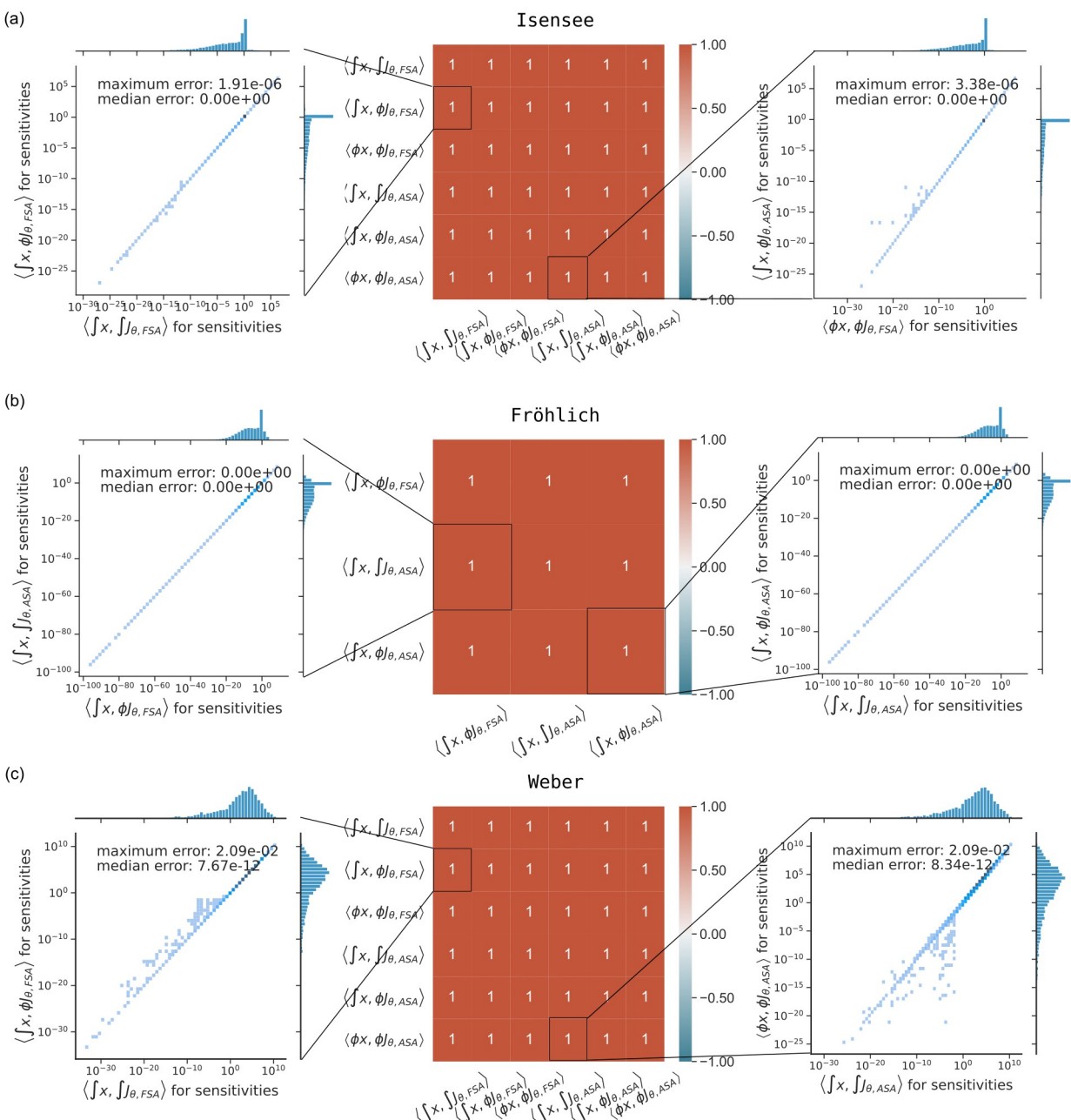

**Fig 3. Comparison of objective function gradients obtained from different method pairs.** The heatmaps show Pearson correlation coefficients between objective function gradient values computed with the six different method pairs. Data was log-transformed prior to the computation of the coefficients. Scatter plots visualize the difference between objective function gradient values for selected method pairs. Points on the diagonal indicate a good agreement, darker points indicate higher density. The maximum and median deviations were computed as defined Section "Accuracy of gradient computation" of the S1 Text.

`Isensee`, `Weber` and `Zheng` models. For example, for the `Weber` model the pre-equilibration speedup of $\langle \int x, \phi \, \mathcal{J}_{\theta, \mathrm{FSA}} \rangle$ over $\langle \int x, \int \mathcal{J}_{\theta, \mathrm{FSA}} \rangle$ is 28.7, while the total simulation time speedup for the same method pairs is 4.2.

In most cases, using Newton's method instead of numerical integration to compute steady states, as well as using tailored sensitivity-at-steady-state methods, reduced simulation time.

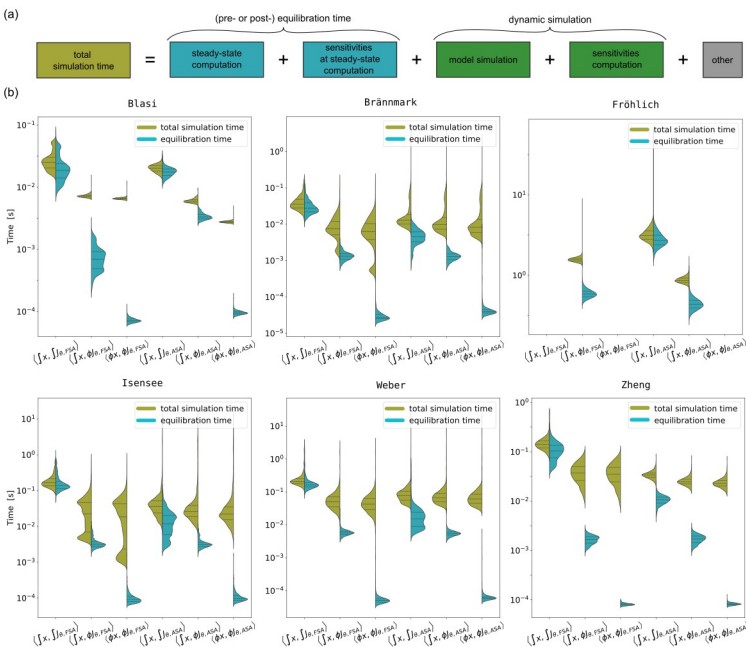

**Fig 4. Comparison of total simulation times and equilibration times for different combinations of problems and steady-state sensitivity methods.** (a) Composition of total simulation time. The main components are, depending on the model, pre- or post-equilibration including sensitivities computation and dynamic simulation (only for `Brännmark`, `Isensee`, `Weber` and `Zheng` models). (b) Violin plots comparing simulation efficiency of the method pairs. The left side on the violin plots shows total simulation time, the right side—total equilibration time.

Speedups of total simulation time ranged from 3.5 to 7.3 for $\langle \int x, \phi\, \mathcal{J}_{\theta,\mathrm{FSA}}\rangle$ over $\langle \int x, \int \mathcal{J}_{\theta,\mathrm{FSA}}\rangle$ (1.2 to 3.6 for $\langle \int x, \phi\, \mathcal{J}_{\theta,\mathrm{ASA}}\rangle$ over $\langle \int x, \int \mathcal{J}_{\theta,\mathrm{ASA}}\rangle$), from 1.05 to 1.2 for $\langle \phi x, \phi\, \mathcal{J}_{\theta,\mathrm{FSA}}\rangle$ over $\langle \int x, \phi\, \mathcal{J}_{\theta,\mathrm{FSA}}\rangle$ (1 to 2.2 for $\langle \phi x, \phi\, \mathcal{J}_{\theta,\mathrm{ASA}}\rangle$ over $\langle \int x, \phi\, \mathcal{J}_{\theta,\mathrm{ASA}}\rangle$). Our analysis shows that, while $\langle \int x, \int \mathcal{J}_{\theta,\mathrm{FSA}}\rangle$ was the slowest approach for all problems, the fastest approach was model-dependent. For all models (except for the `Fröhlich` model) the fastest method pair was either $\langle \phi x, \phi\, \mathcal{J}_{\theta,\mathrm{FSA}}\rangle$ or $\langle \phi x, \phi\, \mathcal{J}_{\theta,\mathrm{ASA}}\rangle$.

## Using sensitivity-at-steady-state methods significantly reduces optimization time

Simulation efficiency is especially important when large numbers of simulations have to be performed. This is often the case during parameter estimation where thousands to millions of objective function evaluations, and thus, model simulations are required. Accordingly, total optimization time is comprised of the cumulative total simulation time over all optimization steps, the time taken by the optimizer, and any additional overhead (Fig 5a).

Therefore, to assess how the choice of steady-state and sensitivity-at-steady-state methods affects optimization efficiency and robustness, we ran 1000 local optimizations for the `Blasi`, `Brännmark`, `Isensee`, `Weber` and `Zheng` models for each method pair. With the `Fröhlich` model, only 50 local optimizations were performed with a maximum of 30 optimizer steps, due to the need for massive computational resources. Furthermore, for the `Fröhlich` model we disregarded $\langle \phi x, \phi\, \mathcal{J}_{\theta,\mathrm{FSA}}\rangle$ and $\langle \phi x, \phi\, \mathcal{J}_{\theta,\mathrm{ASA}}\rangle$ due to high failure rates and $\langle \int x, \int \mathcal{J}_{\theta,\mathrm{FSA}}\rangle$ due to inefficiency.

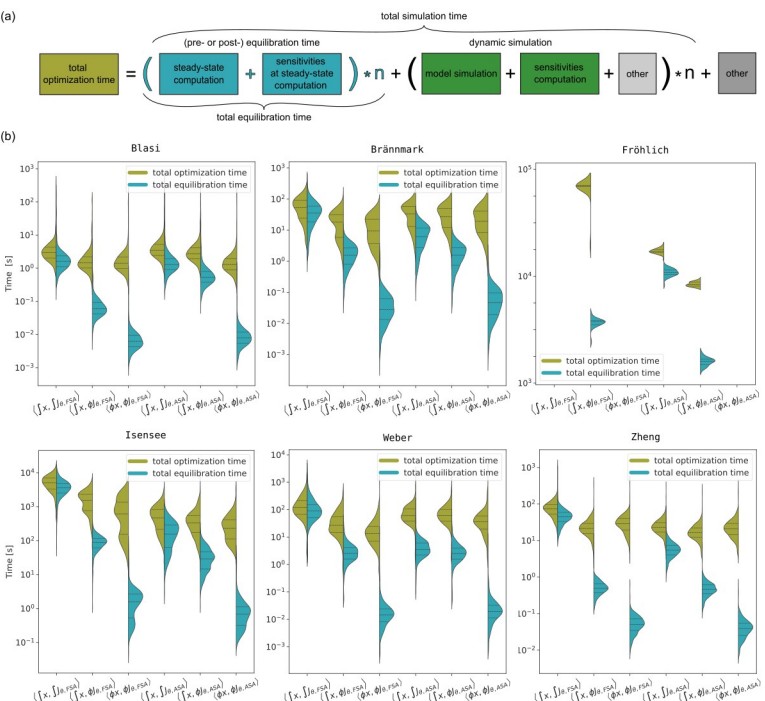

**Fig 5. Optimization efficiency, total time *vs*. equilibration time.** (a) Composition of total optimization time, where *n* is the number of optimization steps. Multiple simulations are required during an optimization, each, depending on the model, might include equilibration and dynamic simulation. (b) Violin plots comparing optimization efficiency of the method pairs.

The assessment of the optimization results revealed that the number of optimization failures correlated with the number of simulation failures (S5 Fig in S1 Text) and all method pairs had comparable efficacy (S6 Fig in S1 Text). Therefore, we focused on comparing the method pairs efficiency. The speedups of $\langle \phi x, \phi \mathcal{J}_{\theta,\mathrm{FSA}} \rangle$ compared to $\langle \int x, \int \mathcal{J}_{\theta,\mathrm{FSA}} \rangle$ were 2.1, 5.6, 8.2, 8.6, 2.6 for Blasi, Brännmark, Isensee, Weber and Zheng models, respectively, and the speedups of $\langle \phi x, \phi \mathcal{J}_{\theta,\mathrm{ASA}} \rangle$ compared to $\langle \int x, \int \mathcal{J}_{\theta,\mathrm{ASA}} \rangle$ were 2.7, 1.7, 2.0, 1.7, 1.1 for Blasi, Brännmark, Isensee, Weber and Zheng models, respectively. The speedups of $\langle \int x, \phi \mathcal{J}_{\theta,\mathrm{FSA}} \rangle$ compared to $\langle \int x, \int \mathcal{J}_{\theta,\mathrm{FSA}} \rangle$ were 2.0, 2.9, 3.3, 4.4, 3.5 for Blasi, Brännmark, Isensee, Weber and Zheng models, respectively, and $\langle \int x, \phi \mathcal{J}_{\theta,\mathrm{ASA}} \rangle$ compared to $\langle \int x, \int \mathcal{J}_{\theta,\mathrm{ASA}} \rangle$ was 1.3, 1.2, 1.4, 2.0, 1.0, 1.4 times as fast for the Blasi, Brännmark, Isensee, Fröhlich, Weber and Zheng models, respectively.

Overall, as with total simulation time, we observed that both using Newton's method instead of numerical integration, and using tailored sensitivities-at-steady-state approaches, substantially reduces total equilibration time, i.e. cumulative equilibration time over all optimization steps, (Fig 5b, right half of the violin plots). The effect on total optimization time is not as pronounced, because the difference in methods is only in relation to simulation time, particularly equilibration time, but not the time taken in the optimizer itself, or other overhead. For most model, the fastest method pair was either $\langle \phi x, \phi \mathcal{J}_{\theta,\mathrm{FSA}} \rangle$ and $\langle \phi x, \phi \mathcal{J}_{\theta,\mathrm{ASA}} \rangle$. One should keep in mind, however, that for some models these method pairs may not be applicable or result in many multi-start failures at the starting point. Fortunately, the more robust method pairs $\langle \int x, \phi \mathcal{J}_{\theta,\mathrm{FSA}} \rangle$ and $\langle \int x, \phi \mathcal{J}_{\theta,\mathrm{ASA}} \rangle$ are more efficient compared to $\langle \int x, \int \mathcal{J}_{\theta,\mathrm{FSA}} \rangle$ and $\langle \int x, \int \mathcal{J}_{\theta,\mathrm{ASA}} \rangle$.

## Discussion and conclusion

In this study, we explored six combinations of methods for computing steady states and sensitivities at steady state, and evaluated their accuracy, efficiency and robustness based on six real-world problems. All those methods have been introduced before, but have not yet been evaluated on a joint set of problems [1, 10, 12, 17]. The considered methods are widely applicable and do not require any specific model properties. However, it should be taken into account that the standard FSA approach is the least scalable and becomes prohibitively computationally expensive for large-scale models, while the more scalable ASA approach avoids computation of state sensitivities that may be needed for further analysis. Our results indicate that tailored sensitivities computation methods are more robust and more efficient than the standard FSA or ASA approaches that rely on numerical integration. In contrast, we found that using numerical integration for steady-state computation was more robust than using Newton's method. However, when Newton's method worked reliably, it was more than twice as fast.

An important issue we encountered was that computing steady states using Newton's method led to non-physical solutions, in our case negative concentrations. This is a general problem with Newton's method and alternative approaches able to respect non-negativity constraints, or better general constraints on the steady-state, while maintaining much of the efficiency would be valuable. However, we are not aware of any such implementation in the context of ODE model simulators.

Different subsets of the methods evaluated here have also been investigated in earlier studies [10–12]. Using Newton's method for steady-state computation was demonstrated and evaluated in [12]. While we also observed a speedup when using Newton's method, we found it only to be roughly twice as fast, instead of 100 times as fast as in the earlier study for the same model. This might be explained best by the comparatively small distances of the initial states from the steady state in [12]. The high failure rate we observed for the Fröhlich model is in line with the earlier observations. The tailored ASA method for computation of steady-state sensitivities ($\langle \int x, \phi \, \mathcal{J}_{\theta,\mathrm{ASA}} \rangle$) was introduced and compared to the standard ASA approach ($\langle \int x, \int \mathcal{J}_{\theta,\mathrm{ASA}} \rangle$) in [10]. Simulation and optimization speedup values we observed in this study are in line with the previous results for the same models.

While we would like to derive general guidelines for choosing the best method for a given model, we have to note that our results might not generalize fully. Our study showed that the impact of the explored approaches on failure rates and speedup are quite problem-dependent. We based our analysis on six of the seven models available in the PEtab benchmark collection [9]. Further extension of the model collection, in particular by models that require steady-state computation, would allow for a more comprehensive analysis. Furthermore, much of the robustness and efficiency will depend on the exact implementation of the different algorithms and the chosen hyperparameters. It has been demonstrated elsewhere in the context of optimization, how different implementations of the supposedly same algorithm can perform quite differently [27]. Our results are based on AMICI [26]. We were not able to find another tool that would have allowed us to compare all the different methods explored here.

In summary, we illustrated different approaches for computing sensitivities for ODE simulations involving steady-states and demonstrated their advantages and disadvantages. While we focused here on systems biology applications, we think that these results mostly transfer to other fields, and can give modelers at least a first orientation on which methods to choose. Our comparison suggests one should be cautious when using Newton's method for steady-state computation, as it might lead to a high number of failures or unphysical solutions. On the other hand, using tailored sensitivity-at-steady-state methods ($\langle \int x, \phi \, \mathcal{J}_{\theta,\mathrm{FSA}} \rangle$ or

$\langle \int x, \phi\, \mathcal{J}_{\theta,\mathrm{ASA}} \rangle)$ is advantageous as these methods are robust and lead to a significant computational speedup.

## Supporting information

**S1 Text. Supplementary notes on accuracy assessment, implementation details, conserved quantities, as well as supplementary figures.**
(PDF)

## Author Contributions

**Conceptualization:** Polina Lakrisenko, Dilan Pathirana, Daniel Weindl, Jan Hasenauer.

**Formal analysis:** Polina Lakrisenko.

**Funding acquisition:** Daniel Weindl, Jan Hasenauer.

**Investigation:** Polina Lakrisenko.

**Methodology:** Polina Lakrisenko, Dilan Pathirana, Daniel Weindl, Jan Hasenauer.

**Supervision:** Dilan Pathirana, Daniel Weindl, Jan Hasenauer.

**Visualization:** Polina Lakrisenko.

**Writing – original draft:** Polina Lakrisenko.

**Writing – review & editing:** Polina Lakrisenko, Dilan Pathirana, Daniel Weindl, Jan Hasenauer.

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
