## [Decision Letter · Decision Letter 0]

6 Aug 2024

PONE-D-24-23601Exploration of methods for computing sensitivities in ODE models at dynamic and steady statesPLOS ONE

Dear Dr. Lakrisenko,

Thank you for submitting your manuscript to PLOS ONE. After careful consideration, we feel that it has merit but does not fully meet PLOS ONE’s publication criteria as it currently stands. Therefore, we invite you to submit a revised version of the manuscript that addresses the points raised during the review process.

**ACADEMIC EDITOR: ** The manuscript has scholarly importance, and it seems very interesting. However, the title of this manuscript appears to need improvement. Try to improve your manuscript title, abstract, and introduction. The abstract should be aligned or centered. "Experimenting with methods for computing sensitivities in ordinary differential equation models" would have been fine. Other comments(1) ODE should be defined as ordinary differential equation before being used. (2) Missing punctuation mark in equation (8).(3) The authors are not consistent with a particular referencing style. Thus, the authors should ensure that they format the references according to PLOS ONE’s standards. (4) Are the figures part of the main work? If they are, then they should be included in the main manuscript and should be explicated. They should be given figure numbers. The first pdf figures is not clear.(6) Authors should apply these 6 methods to a given ODE model and compare the

numerical results at pre-equilibration and post-equilibration(7) What is the difference between this current work and " PeTTSy: a computational tool for perturbation analysis of complex systems biology models" and Sensitivity analysis of infectious disease models: methods, advances and their application  

We look forward to receiving your revised manuscript.

Kind regards,

Joshua Kiddy K. Asamoah, PhD

Academic Editor

PLOS ONE

Journal Requirements:

2. Thank you for stating the following financial disclosure: "This work was supported by the Deutsche Forschungsgemeinschaft (DFG, German Research Foundation, https://www.dfg.de/) under Germany’s Excellence Strategy (EXC 2047—390685813, EXC 2151—390873048; J.H.) and under the project IDs 432325352 – SFB 1454 and 443187771 – AMICI (D.P.), by the German Federal Ministry of Education and Research (BMBF, https://www.bmbf.de) within the e:Med funding scheme (junior research alliance PeriNAA, grant no. 01ZX1916A; D.W., P.L.), and by the University of Bonn (https://www.uni-bonn.de, via the Schlegel Professorship of J.H.)." 

Reviewers' comments:

Reviewer's Responses to Questions

**Comments to the Author**

1. Is the manuscript technically sound, and do the data support the conclusions?

Reviewer #1: Yes

Reviewer #2: Yes

2. Has the statistical analysis been performed appropriately and rigorously? 

Reviewer #1: No

Reviewer #2: N/A

3. Have the authors made all data underlying the findings in their manuscript fully available?

Reviewer #1: Yes

Reviewer #2: Yes

4. Is the manuscript presented in an intelligible fashion and written in standard English?

Reviewer #1: Yes

Reviewer #2: Yes

5. Review Comments to the Author

Reviewer #1: Comments:

1. The abstract and the introduction look well written.

2. Are these 6 methods analyzed in this paper new? What is the new content in

this article? The authors compared 6 different methods to solve a steady-state and

sensitivities analysis for an ODE model.

3. The sentence on lines 54 to 63, does this add any new approach to science since

these methods exist?

4. The Figures are blurry, therefore making the review process very difficult

5. Authors should apply these 6 methods to a given ODE model and compare the

numerical results at pre-equilibration and post-equilibration

6. In conclusion, discuss your results’ result/weakness/applicability. The results need

to be elaborated.

1

See pdf

Reviewer #2: The study explore six method pairs for computing the steady state and sensitivities at steady state using six real-world problems. The method pairs involve numerical integration or Newton’s method to compute the steady-state, and – for both forward and adjoint sensitivity analysis–numerical integration or a tailored method to compute the sensitivities at steady-state. The paper is potentially interesting and useful. However, the author needs to pay attention to the following suggestions:

#The abstract general overview is too long and lacks specific results that highlight the main findings. It would be beneficial to include a sentence summarising the key outcomes of this study.

#Consider increasing the visibility of the figures and give proper citation for each in the discussion.

#Provide more detailed explanations of failure cases, particularly for methods that showed high failure rates (e.g., Newton’s method for the Fröhlich model).

6. PLOS authors have the option to publish the peer review history of their article (what does this mean?). If published, this will include your full peer review and any attached files.

Reviewer #1: **Yes: **Chidozie Willaims Chukwu

Reviewer #2: **Yes: **Emmanuel Addai

---

## [Author Response · Author response to Decision Letter 0]

19 Sep 2024

Dear Dr Joshua Kiddy K. Asamoah,

Dear Dr Chidozie Willaims Chukwu,

Dear Dr Emmanuel Addai,

We thank you for your constructive comments on our manuscript “Exploration of methods for

computing sensitivities in ODE models at dynamic and steady states”. Based on your remarks,

we have revised the manuscript. Our point-by-point response is in the "Response to Reviewers.pdf" file.

---

## [Decision Letter · Decision Letter 1]

2 Oct 2024

Benchmarking methods for computing local sensitivities in ordinary differential equation models at dynamic and steady states

PONE-D-24-23601R1

Dear Dr. Lakrisenko,

We’re pleased to inform you that your manuscript has been judged scientifically suitable for publication and will be formally accepted for publication once it meets all outstanding technical requirements.

Kind regards,

Joshua Kiddy K. Asamoah, PhD

Academic Editor

PLOS ONE

Additional Editor Comments (optional):

One reviewer accepted you paper while the second did not accept our invitation to review the paper again. I have carefully checked your responses to his comments and am satisfied with the correction and modification.

Reviewers' comments:

Reviewer's Responses to Questions

**Comments to the Author**

1. If the authors have adequately addressed your comments raised in a previous round of review and you feel that this manuscript is now acceptable for publication, you may indicate that here to bypass the “Comments to the Author” section, enter your conflict of interest statement in the “Confidential to Editor” section, and submit your "Accept" recommendation.

Reviewer #2: All comments have been addressed

2. Is the manuscript technically sound, and do the data support the conclusions?

Reviewer #2: Yes

3. Has the statistical analysis been performed appropriately and rigorously? 

Reviewer #2: Yes

4. Have the authors made all data underlying the findings in their manuscript fully available?

Reviewer #2: Yes

5. Is the manuscript presented in an intelligible fashion and written in standard English?

Reviewer #2: Yes

6. Review Comments to the Author

Reviewer #2: I have no further comments. I think Editor can make decision of this manuscript. All comments have been address accordingly

7. PLOS authors have the option to publish the peer review history of their article (what does this mean?). If published, this will include your full peer review and any attached files.

Reviewer #2: **Yes: **Emmanuel Addai

---

## [Editor Report · Acceptance letter]

9 Oct 2024

PONE-D-24-23601R1 

PLOS ONE

Dear Dr. Lakrisenko, 

I'm pleased to inform you that your manuscript has been deemed suitable for publication in PLOS ONE. Congratulations! Your manuscript is now being handed over to our production team.

Kind regards, 

on behalf of

Dr. Joshua Kiddy K. Asamoah 

Academic Editor

PLOS ONE